# 3D Device for Forces in Swimming Starts and Turns

**Karla de Jesus** [1,2,3,4,*], **Luis Mourão** [1,2,5], **Hélio Roesler** [6], **Nuno Viriato** [2,7],
**Kelly de Jesus** [1,2,3,4], **Mário Vaz** [2,7], **Ricardo Fernandes** [1,2] and **João Paulo Vilas-Boas** [1,2]

1   Centre of Research, Education, Innovation and Intervention in Sport, Faculty of Sport, University of Porto, 91 Dr. Plácido Costa st., 4200-450 Porto, Portugal
2   Porto Biomechanics Laboratory, University of Porto, 91 Dr. Plácido Costa st., 4200-450 Porto, Portugal
3   Human Performance Studies Laboratory, Faculty of Physical Education and Physiotherapy, Federal University of Amazonas, 3000 Gal. Rodrigo Octávio Jordão Ramos ave., South MiniCampus, Coroado I, Manaus 69077-000, Amazonas, Brazil
4   Human Motor Behaviour Studies Laboratory, Faculty of Physical Education and Physiotherapy, Federal University of Amazonas. 3000 Gal. Rodrigo Octávio Jordão Ramos ave., South MiniCampus, Coroado I, Manaus 69077-000, Amazonas, Brazil
5   Superior Institute of Engineering of Porto, Polytechnic Institute of Porto, 431, Dr. António Bernardino de Almeida st., 4249-015 Porto, Portugal
6   Centre of Physical Education, Physiotherapy and Sports, Aquatic Biomechanics Research Laboratory, University of the State of Santa Catarina, 358 Pascoal Simone st., Coqueiros, Florianópolis 88080-350, Santa Catarina, Brazil
7   Institute of Mechanical Engineering and Industrial Management, Faculty of Engineering, University of Porto, Dr. Roberto Frias st., 4200-465 Porto, Portugal
*   Correspondence: karladejesus@ufam.edu.br; Tel.: +55-92-3305-1181

**Abstract:** Biomechanical tools capable of detecting external forces in swimming starts and turns have been developed since 1970. This study described the development and validation of a three-dimensional (six-degrees of freedom) instrumented block for swimming starts and turns. Seven force plates, a starting block, an underwater structure, one pair of handgrips and feet supports for starts were firstly designed, numerically simulated, manufactured and validated according to the Fédération Internationale de Natation rules. Static and dynamic force plate simulations revealed deformations between 290 to 376 $\mu\varepsilon$ and 279 to 545 $\mu\varepsilon$ in the anterior-posterior and vertical axis and 182 to 328.6 Hz resonance frequencies. Force plates were instrumented with 24 strain gauges each connected to full Wheatstone bridge circuits. Static and dynamic calibration revealed linearity ($R^2$ between 0.97 and 0.99) and non-meaningful cross-talk between orthogonal (1%) axes. Laboratory and ecological validation revealed the similarity between force curve profiles. The need for discriminating each upper and lower limb force responses has implied a final nine-force plates solution with seven above and two underwater platforms. The instrumented block has given an unprecedented contribution to accurate external force measurements in swimming starts and turns.

**Keywords:** sports engineering; biomechanics; ground reaction forces; swimming; performance

## 1. Introduction

The analysis of swimmers' performance has traditionally examined spatiotemporal variables representing the start, turn and clean swimming distance [1]. The start and turn are fundamentally different skills than free swimming, which does not necessarily indicate a similar level of start or turning performance [1]. Generally, at the elite level, it is not only swimming speed that wins the races but rather the start and turn where most expert swimmers are travelling at their fastest velocity [2,3].

For instance, the start phase of the 50 m men's freestyle at the 2019 FINA Champions Swim Series at Budapest has showed that 15 m after the start take-off phase, the second-placed swimmer was 0.08 s slower and the final race time difference was 0.15 s. Moreover, over a 200 m event, the turn contributes 21% to total race performance and progressively more as race distance increases [4].

External forces generated during the ventral starts using instrumented starting blocks have been measured since 1971 with a uniaxial force platform placed at the edge of the pool, which had assessed horizontal forces applied by the swimmers' feet during the conventional circular arm swing technique [5]. Cavanagh et al. [6] had optimised Elliot and Sinclair's solution including eight strain gauges in a horizontal bar to measure the horizontal and vertical forces applied by the hands and feet during the grab ventral start technique. Three-dimensional forces were assessed in starts since 2003 by Naemi et al. [7] revealing that the grab start performed with hands in between feet was less stable in preventing platform twist. Horizontal forces applied by feet and hands in the backstroke start were firstly measured in 2011 (de Jesus) using a uniaxial platform and a load cell, being fixed on the starting wall and on the handgrip, respectively.

The changes in the starting block design in 2008 implied adjustments in the previously developed instrumented platforms for start (e.g., [2]) force analyses. Mason and co-authors [2] have presented an instrumented block comprised of four triaxial force sensors placed in a main force plate fixed over the starting block, inclined rear plate, handles for ventral and backstroke start and underwater platform with holes over the front surface. Recently, researchers have shown a new device to measure independently three-axial forces exerted on hands and feet in the ventral kick start technique [8], but limitations have been identified on hand placement dependence. Despite the relevant shortcomings for start analysis that the previous systems have provided, none of them are able to narrow in a unique solution for all ventral, backstroke and relay start force analysis possibilities, assuming laterality effects and independency in hands and feet force assessment. The horizontal and vertical force components determine steering start strategies, being the most assessed. However, lateral responses are essentially a controlling movement in starts [7,9,10] and it can be better understood when assessing each swimmer's limb force contribution.

The description of forces generated during swimming turns also started in the 1970s using uniaxial force plates to assess the horizontal component in tumble and open turns (e.g., [11]). Swimming turns external forces analysis using three-dimensional force plates is still scarce, with the flip technique and its variants being the most commonly investigated (e.g., [12,13]). A double underwater tri-axial force plate solution developed for independent assessment of forces applied by each foot in a backstroke start can also provide a direct measurement of the combined forces exerted during turns and the determination of foot position. Furthermore, in some turn techniques, coaches can assess laterality data if the foot location is compatible with each force platform (e.g., [13]). Mason and co-authors' underwater force platform design has a multitude of holes system to reduce wave effects [2]. However, this configuration limits the possibility to use the same force plate for water wave effect analyses [14].

Coaching and commercial ongoing-instrumented starting blocks are still lacking some final integrated solution for start and turn force analyses that could inspire new biomechanical research directions. In a competitive, rapid uptake market such as sports equipment, it is important to keep searching for new and improved designs and materials at affordable prices [15]. Commercial force plate prices are ~$20,000, which can usually be very expensive for coaches and biomechanists [16]. Beside the high prices, technical assistance issues and spare parts' availability are sources of many difficulties for labs and researchers [17]. Thus, due to the high cost of existing force plate systems, the development of simpler low cost, adjustable and accurate models for biomechanical analysis is desirable for force measurements in swimming start and turn techniques (cf. [18]).

A dynamometric unit composed by independent 3D force plates, a starting block, an underwater structure, ventral and dorsal start handgrips and feet supports is original. It presents crucial potentialities, particularly regarding forces and momenta measurement in individual ventral and backstroke starts, relays and turning techniques. As a versatile and low cost system, the force

plates can be used uncoupled from the dynamometric unit to measure active (e.g., [19]) and passive swimmers' drag (e.g., [14]), and underwater gait ground reaction forces (e.g., [20]). The current study aimed to design, construct and validate an instrumented swimming start block, emphasizing geometry description, numerical simulation, sensors bonding, calibration, experimental and ecological validation procedures.

## 2. Materials and Methods

### 2.1. 3D Geometric Computer Aided Design

Force plates, the starting block, underwater structure, handgrips and feet supports were 3D designed using a solid modelling computer aided design software (SolidWorks 2012, Dassault Systèmes, SOLIDWORKS Corporation, Waltham, MA, USA). Each force plate and handgrip was framed to achieve proper sensibility with a high rigidity and reduced mass. The start block project prioritized low deformation to support seven to nine force plates, handgrips and an underwater structure. In addition, force plates, the start block, underwater structure, handgrips and feet support dimensions (Figure 1) were conditioned to comply with the Fédération Internationale de Natation rules (FINA; FR 2.7 and FR 2.10).

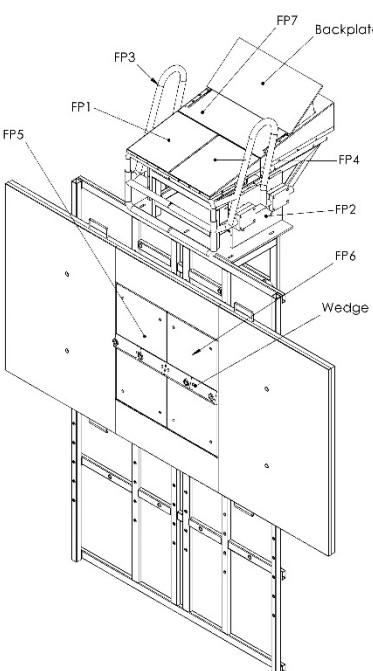

**Figure 1.** The seven force plates solution distributed in the start block for individual limb force-moment assessments: Each force plate (FP 1 to 7) and feet supports (back plate and wedge) are also shown.

### 2.2. Force Plates Spatial Layout

Contrarily to arbitrarily multiplying force platforms on the block, it is worth it to consider a minimum number of them. Such a number considers the limbs involved in the starting technique force development and the set of limb positioning on the starting block/wall. The rationale relies on the following: If a sole limb force is measured then the centre of pressure ($\overrightarrow{COP}$) lies on the limb contact area. On the other hand, if a sole platform measures simultaneously two-limb forces then the centre of pressure lies between the limbs and perhaps outside each limb contact area. Such measurement can mask postural constraints necessary for maximum performance data interpretation. The five above water force plates configuration was firstly developed for independent swimmers' upper and lower limbs kinetic analysis in individual ventral start techniques (e.g., grab and track start).

Subsets of analogous platforms have been preferable, aiming for the comparison of force plates behaviour, which implied that two pairs of platforms were similar (300 mm × 250 mm above water and 600 mm × 300 mm underwater). This resemblance provides an interchangeable ability, which might be of great importance if mass production or maintenance were considered. The fifth above water force plate is 500 mm × 440 mm, enabling support for the rear foot as used in the track start technique.

### 2.3. Force Plates Geometry

The main requirements of the load sensor followed Lywood et al.'s geometry [21], which consisted of instrumented rectangular section bars oriented to be most sensitive to $x$, $y$ and $z$ force and torque components (cf. [22]), obviating most cross-talk. A waterproof sensor choice enables force plates immersion into the bottom of the swimming pool (~2 m). Sensor locus, distant from the centre anchorage points, allows a better interpolation approach. As the Roesler [20] platform design complied with these previous requirements, it was selected to serve as the testing tool.

It has already been mentioned that Roesler's [20] force plate topology allows a more accurate $\overrightarrow{COP}$ determination, as well as direct and independent 3D forces and moment measurements without interference among them [18]. Each force plate core was designed to be manufactured in galvanized steel and is essentially composed by two vertical and two horizontal beams plus two lateral boxes. The beams, contrarily to the ring or pylon, can acquire the applied load with better accuracy and precision, also allowing better minute change capture in the strain throughout the top plate and not just at the corners [23]. Above and underwater top (Figure 2a,b) and bottom (Figure 2c,d) force plates were built in duralumin to minimise force plate mass. The mounting apparatus plays a crucial role in providing accurate and reliable measurements, with the force plate top and core unattached with commercial bushings.

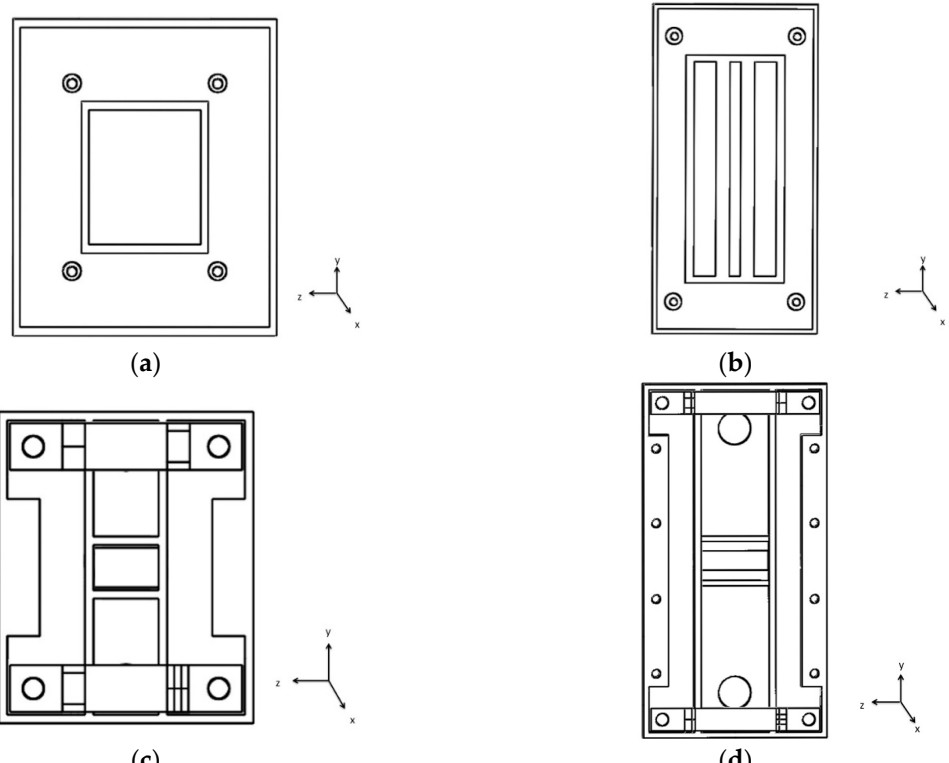

**Figure 2.** Above and underwater top (**a**,**b** panels) and core (**c**,**d** panels) 300 mm × 250 mm and 600 mm × 300 mm force plates top.

### 2.4. Start Block, Underwater Structure, Handgrips and Feet Support

Two start block projects were previously designed: (i) a bulky and solid structure and (ii) a lattice block with a declination support, which was replaced by a lattice galvanized steel structure with zero inclination due to excessive mass and reduced stability showed by the previous structures. The starting block was projected to be fixed over an underwater structure attached vertically to the swimming pool wall by front and rear edges, being similar to a previous structure used by Pereira et al. [13,14] and de Jesus et al. [24] for flip turn and backstroke start kinetics assessment. The first underwater structure included a two independent force plates support. The underwater structure evolved from a heavier to a lattice form, with its version being slighter, including holes with 100 mm distance between them placed at different heights on both force plates. It presented a flat rectangular surface for swimming turn analysis (replying the touch pad; FINA, FR 2.13), with a hollow area for underwater force plates embedding, as previously used in turn analysis [13,14].

Handgrips were projected to be independent and framed in galvanized steel, being the first design very versatile to be easily used in ventral and dorsal start technique analysis. However, the first design had shown a handgrip positioning dependency on measured strain signals that did not allow real training and competitive swimmers' movement. In fact, strain gauges would be bonded to the handgrip pipes, obliging swimmers to position their hands on a fixed place to allow comparisons (this was eliminated by the handgrips fixation on each lateral force plate top). With two force plates fixed, each one on the start block lateral, forces and moments could be measured and handgrips positioning could be located. The second handgrip project was based on a simple and outdated handgrip version (e.g., [24]), being updated for two horizontal (i.e., the highest and the lowest, 0.43 and 0.56 m above water surface, respectively) and vertical bars, following the OSB11 starting block configurations. The final handgrip prototype received fine arrangements due to the existent pipe profile. The adjustable feet support for ventral and backstroke start was framed in galvanised steel and nylon (FINA, FR 2.7 and 2.10, respectively), allowing the five rear foot authorised positions.

### 2.5. Finite Element Analysis

We conducted static structural simulations using modelling software for finite element analysis (Ansys v.12.1, ANSYS Inc., Canonsburg, PA, USA), thus enabling predictions on how the dynamometric unit would strain under isolated and integrated conditions. Dynamic simulations were applied to verify resonance frequency, equivalent stress, equivalent strain and deformations. Based on Roesler et al. [20] geometry and sensor location definition, a 8000 N load was vertically and antero-posteriorly applied to confirm the previously determined sensor location (Figure 3a). The 8000 N load was simulated to allow force plate use in other data collection purposes that had depicted increased ground reaction forces (e.g., long distance jump, 15.2 times body weight; [25]).

Static and dynamic force plate simulations were performed with a core, top and mounting apparatus (i.e., polyethylene bushing and screw). Simulations with the starting block and handgrips were conducted with a 2500 N (centrally located; Figure 3b) and 2000 N load (vertical and antero-posterior; Figure 3c) in the commonly used handgrip positioning for ventral and backstroke start [10]. The underwater structure with force plates vertically mounted was simulated with the gravity at sea level (i.e., 9.81 m/s$^2$) and values of total deformation, equivalent stress and equivalent elastic strain have also been obtained. The most refined mesh for force plates, the starting block, underwater structures and handgrip simulations was composed of pyramids and cobbled with a 1 mm length and 39,197 nodes and 12,431 elements, 76,344 nodes and 15,724 elements, 273,837 nodes and 107,653 elements, and 46,372 nodes and 17,459 elements, respectively.

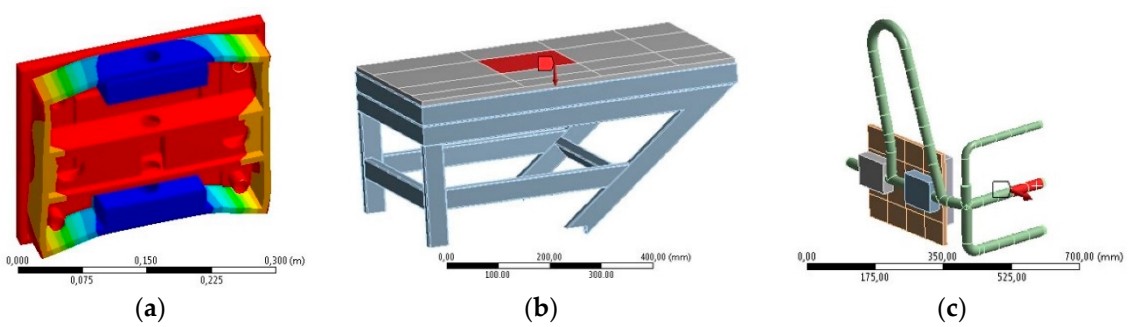

**Figure 3.** Strain numerical simulation in the assembled force plate (**a** panel), starting block (**b** panel), and handgrips (**c** panel).

## 2.6. Electrical Circuit

Following structures manufacturing, strain gauges were bonded to the platforms as sensing elements due to the previous research group background and short budget available. Each force plate was instrumented with 24 waterproof strain gauges (Kyowa, Electronic Instruments, KFW-5-120-C1-5M2B, Tokyo, Japan), arranged in six independent full Wheatstone bridges, minimising temperature effects. Those strain gauges were internally bonded to each force plate core and positioned as depicted in Figure 4a–d.

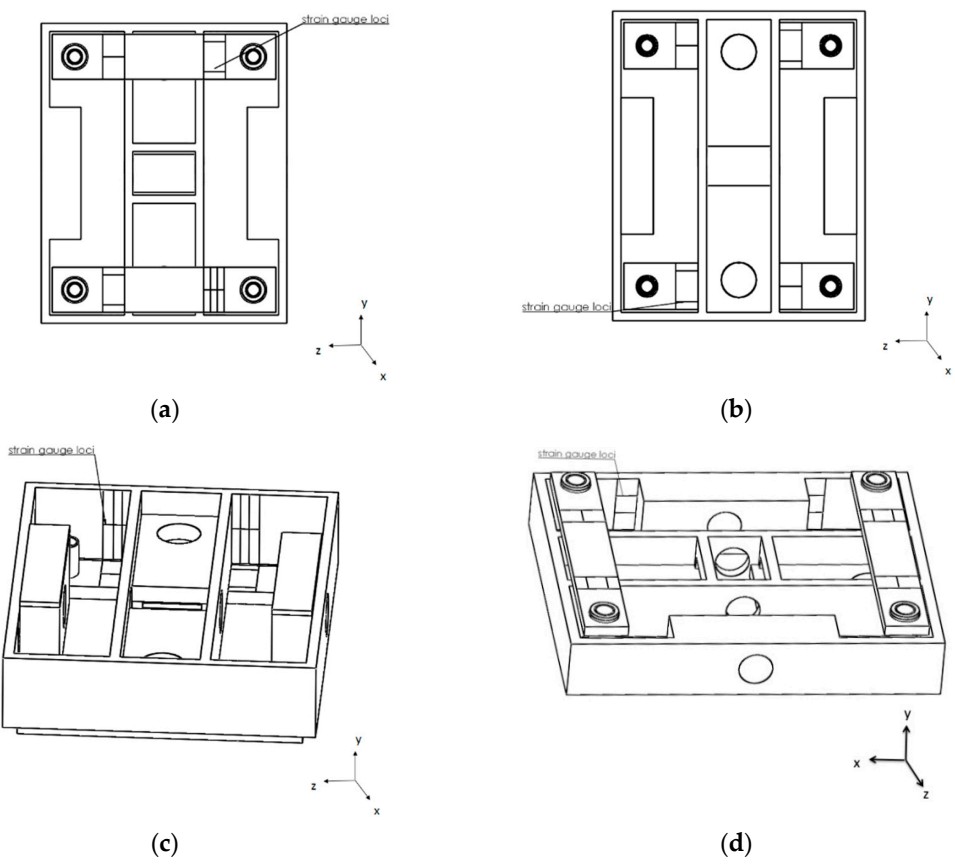

**Figure 4.** Strain gauge zones: Vertical force (y), antero-posterior (x) and medium-lateral (z) moment (top and bottom view, **a**,**b** panel), antero-posterior (x) force (**c** panel) and medium-lateral (z) force and vertical moment (**d** panel).

After bonding, each strain gauge received an additional protection of a two-part polybutadiene resin designed for re-enterable splice protection (Scotchcast TM re-enterable electrical insulation resin

2123, 3M TM, St. Paul, MN, USA), minimising chlorine wear. Strain gauge wires were brazed in a full Wheatstone bridge configuration (six for each force plate), silicone protected. Each full Wheatstone bridge was connected to a shielded and unfilled cable and provided data from each variable of interest (i.e., horizontal, vertical and lateral forces and momenta).

The six-shielded unfilled cables were connected to an analogue-to-digital converter module to transmit full Wheatstone bridge signals (NI9237 50 kS/s/ch 24 bit-4Channels, National Instruments Corporation, NI™, Austin, TX, USA) and to its scanner chassis (NI CompactDAQ 9172 and 9188 with 8 slots, National Instruments Corporation, NI™, Austin, TX, USA) through RJ50 connectivity, which interfaced with the computer. Strain gauges were identified by digit codes and cables wiring had also been standardised with colour code jacket terminals nearby the RJ50 plug. The dynamometric system specs included 42 strain plus two trigger channels acquiring at a 2000 Hz sampling rate.

Custom-designed data processing software was created in LabView 2013 (SP1, NI™, Austin, TX, USA) to acquire, plot and save the strain readings of each three axial force and moment of force component from each force plate. The executable file was programmed to record data in a total of 8 s (4 s before and after the trigger signal), which was a strategy implemented for the full relay start force acquisitions, anticipatory individual start (e.g., pre-activation) and turn actions (e.g., wave drag). Each force and moment of force curve profile was observed in real data time-acquisition and data acquisition files were later operated to transform strain signals into force-moment (using MatLab R2014a, The MathWorks Incorporated, Natick, MA, USA) routines for the required matricial conversion operations.

### 2.7. Static Calibration

Static calibration was performed on dry land before force plate proper use and applied both to a load and to an unload sequence with 10 kg individual masses (up to 50 kg in each positioning; cf. [8,18]) allowing the correspondence between strain and applied load. The vertical force component was calibrated at five positions with the use of a tension, compression machine (Instron 8804 Servohydraulic Fatigue Testing System, Instron®, Ilinois Tool Works Inc., Norwood, MA, USA) that laid the right load on each force plate top centre. On antero-posterior and lateral axis force calibration, platforms were vertically fixed on the wall and the load was applied on each centre of interest through a stainless-steel cable connection. A stainless-steel cable was fixed through holes made on the lateral of each force plate top (three per edge). Forces and moment were calibrated using the central and lateral holes (respectively).

### 2.8. Laboratory Experimental Validation

An experimental validation was completed comparing the results of a rigid body theoretical free rotation around a pivot point $\overrightarrow{COP}$ fall force pattern [26] and the associated strain signal pattern generated. Rigid body inertia moment of the inverted pendulum had to be assessed previously to know mass distribution to allocate its centre of mass and to calculate the centre of mass to the centre of pressure distance. This assessment implies the mass and geometrical dependency of moment of inertia and dynamic behaviour observed in force patterns generated, obviating the use of any force platform except gravity acceleration knowledge.

### 2.9. Ecological Experimental Validation

The instrumented start block was tested in a 25 m long and 1.90 m deep indoor swimming pool for real data acquisitions, which were then qualitatively compared with starting and turning data previously presented in the literature. All experimental procedures conformed to the requirements stipulated in accordance with The Code of Ethics of the World Medical Association (Declaration of Helsinki) and were approved accordingly by the local research ethics committee. Swimmers and

parents and/or guardians (when participants were under 18 years old) provided informed written consent before data collection.

Nine well-trained, healthy and able-bodied male swimmers (mean and standard deviations: 21.09 ± 5.64 years, stature 1.74 ± 0.05 m, and body mass 71.32 ± 11.03 kg) and eight age-group male swimmers engaged on a regular basis in regional- and national-level competitions (mean ± SD: 12.4 ± 0.6 years old, 155.1 ± 13.6 cm of height, 44.6 ± 10.9 kg of body mass, 14.1 ± 5.3% of body fat, with 3.5 ± 1.4 years of experience in competitive swimming and 3.3 ± 0.7 (2–4) of Tanner maturation scale by self-evaluation) volunteered to participate in the start and turn validation study, respectively.

Swimmers performed a familiarisation period with each start and turn technique studied (cf. [27,28]). For the start protocol, each swimmer randomly performed three maximal 15 m trials of backstroke with vertical handgrips, track and one step relay starts. Swimmers performing the turn protocol were also tested in a set of three trials for the open backstroke to the breaststroke medley turning technique. Trials started and finished from the mid-pool (at 12.5 m from the turning wall) and swimmers were instructed to swim in and out at maximum speed until the 12.5 reference [13,28]. Rest periods of 2 min were observed between each repetition in both protocols.

A start signal complying with the FINA SW 4.2 rule was produced through an official device (StartTime IV acoustic start, Swiss Timing Ltd., Corgémento, Switzerland) and instrumented to simultaneously generate an auditory signal and export a pulse to the force plates with convenient signal conditioning. In the one step relay start protocol and open backstroke to breaststroke turn the starter device (without sound) was triggered 7.5 m before the incoming swimmer had touched the wall.

## 3. Results and Discussion

### 3.1. Simulations

Table 1 presents strain in antero-posterior and vertical axes (8000 N centred) and resonance frequency of the first, third and final prototypes of above water 300 mm × 250 mm, 500 mm × 440 mm and underwater 600 mm × 300 mm platforms, corroborating Roesler's [20] suggestions for strain values between 100 and 500 μ$\varepsilon$. The force plates design optimisation was a compromise among maximum rigidity, minimum mass and high frequency, which allows small deformations, uncoupling, good linearity and low hysteresis (c.f. [17]). The specific force plate application determined proper resonance frequencies and the maximal vertical load was supported. The waterproof force plate used by Roesler [20], with 500 mm × 500 mm framed in galvanized steel, obtained a 35 Hz resonance frequency, which was considered sufficient for underwater applications due to over damping effects. However, in the current project, force plate versatility has been prioritized and values higher than ~140 Hz were required (cf. [20]) as they can be both used independently of the starting block and out of the water. Commercial force platforms with a natural frequency ≥150 Hz have been well accepted for triple jump ground reaction forces assessment (e.g., [25]).

**Table 1.** Antero-posterior and vertical strain and their resonance frequency above 300 mm × 250 mm, 600 mm × 300 mm and 500 mm × 440 mm underwater force plates of first, third and final prototypes.

| Project Evolution | Static and Dynamic Simulations | 300 × 250 mm | 600 × 300 mm | 500 × 440 mm |
|---|---|---|---|---|
| First | Antero-posterior (μ$\varepsilon$) | 323.0 | 470.0 | 297.0 |
| | Vertical (μ$\varepsilon$) | 464.0 | 489.0 | 473.0 |
| | Resonance frequency (Hz) | 235.7 | 153.0 | 140.0 |
| Third | Antero-posterior (μ$\varepsilon$) | 287.0 | 357.0 | 349.0 |
| | Vertical (μ$\varepsilon$) | 545.0 | 521.0 | 511.0 |
| | Resonance frequency (Hz) | 323.7 | 197.0 | 177.0 |
| Final | Antero-posterior (μ$\varepsilon$) | 290.0 | 375.0 | 376.0 |
| | Vertical (μ$\varepsilon$) | 545.0 | 541.0 | 279.0 |
| | Resonance frequency (Hz) | 328.6 | 199.2 | 182.0 |

Table 2 shows results from one example of static simulation from each force plate considering all 24 strain gauge responses, which allow noticing maximal cross-talk less than 5% (cf. [20,21]). In fact, Roesler's [19] findings have shown ~3% of maximal interference among loads when 800 N had been applied. In addition, Lywood et al. [21] reported a cross-talk between orthogonal axes less than 5% for 40 N and 10 N of vertical and horizontal forces applied. The 300 mm × 250 mm force plate laterally positioned on the starting block with fixed handgrip was simulated with a 2000 N load applied vertically on the handgrips, indicating an expectable relevant lateral force (strain gauges 17, 18, 23, 24; cf. Figure 4d). When 8000 N was centrally and vertically applied on a 600 mm × 300 mm force plate, only strain gauges responsible for this measurement responded (i.e., 1 to 12, vertical force, antero-posterior and lateral moment; cf. Figure 4a,b).

**Table 2.** The 24 strain gauge responses when a 2000 N and 8000 N vertical load was applied on 300 mm × 250 mm and 600 mm × 300 mm force plates.

| Strain Gauges | Variable | 300 × 250 mm (με) | 600 × 300 mm (με) |
|---|---|---|---|
| 1 | Lateral moment | −4.54 | 78.13 |
| 2 | Vertical force | 84.66 | 301.17 |
| 3 | Lateral moment | −23.73 | 157.09 |
| 4 | Lateral moment | −149.79 | 108.67 |
| 5 | Vertical force | 200.78 | 316.68 |
| 6 | Lateral moment | 83.29 | 93.08 |
| 7 | Antero-posterior moment | −40.01 | −215.51 |
| 8 | Vertical force | 133.75 | −282.81 |
| 9 | Antero-posterior moment | −74.94 | −205.90 |
| 10 | Antero-posterior moment | 9.51 | −160.94 |
| 11 | Antero-posterior moment | 55.70 | −188.20 |
| 12 | Vertical force | −72.43 | −238.46 |
| 13 | Antero-posterior force | −104.08 | 1.60 |
| 14 | Antero-posterior force | 93.30 | 0.52 |
| 15 | Antero-posterior force | −177.25 | 0.62 |
| 16 | Antero-posterior force | 176.88 | 0.14 |
| 17 | Lateral force | 249.58 | 14.55 |
| 18 | Lateral force | 256.12 | −10.41 |
| 19 | Vertical moment | −26.40 | −7.71 |
| 20 | Vertical moment | −45.34 | 13.11 |
| 21 | Vertical moment | 7.09 | 13.10 |
| 22 | Vertical moment | 33.94 | −8.22 |
| 23 | Lateral force | −222.66 | −10.20 |
| 24 | Lateral force | −264.68 | 15.83 |

The starting block total deformation under a 2500 N centre vertical load was 0.00030553 m. The gravity at sea level (9.81 m/s$^2$) tested over the underwater structure and two force plates vertically fixed on it revealed a maximal deformation of 0.00012322 m. Moreover, using the same standard gravity at sea level, the underwater structure with force plates has showed a maximum of 0.00000813 Pa and 0.0000409 m/m, considering equivalent von-Mises stress and equivalent von-Mises elastic strain, respectively, indicating short stress gradients in the underwater structure regions. An antero-posterior 2000 N load applied both on the lowest and on the highest horizontal and vertical handgrips revealed 200, 165 and 115 maximal με. A vertical 2000 N load applied on the lowest and on the highest horizontal and vertical handgrips indicated 591, 585 and 205 maximal με.

## 3.2. Calibrations

The calibration regression equation for the antero-posterior and medium-lateral axis of 300 mm × 250 mm and 600 × 300 mm force plates is depicted in Figure 5a–d. Results evidenced the previously noticed linearity ($R^2$ ranging between 0.97 and 0.99) and non-meaningful cross-talk

between orthogonal axes (small and negligible; <5%) when quantifying any couple of force plate output signals (cf. [20,21]). Calibration results of 300 mm × 250 mm force plates are depicted considering the forces applied on handgrips positioning previously simulated.

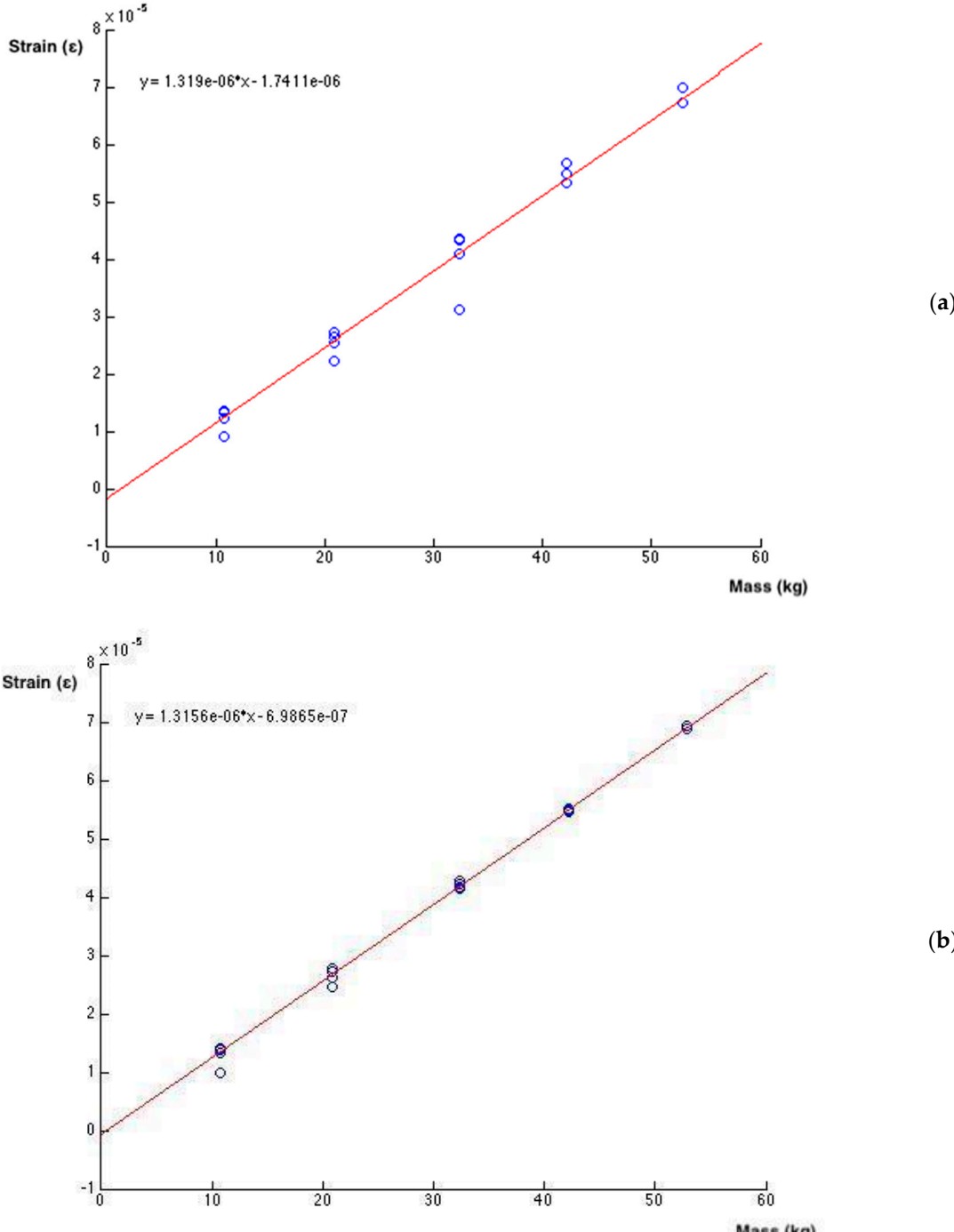

**Figure 5.** *Cont.*

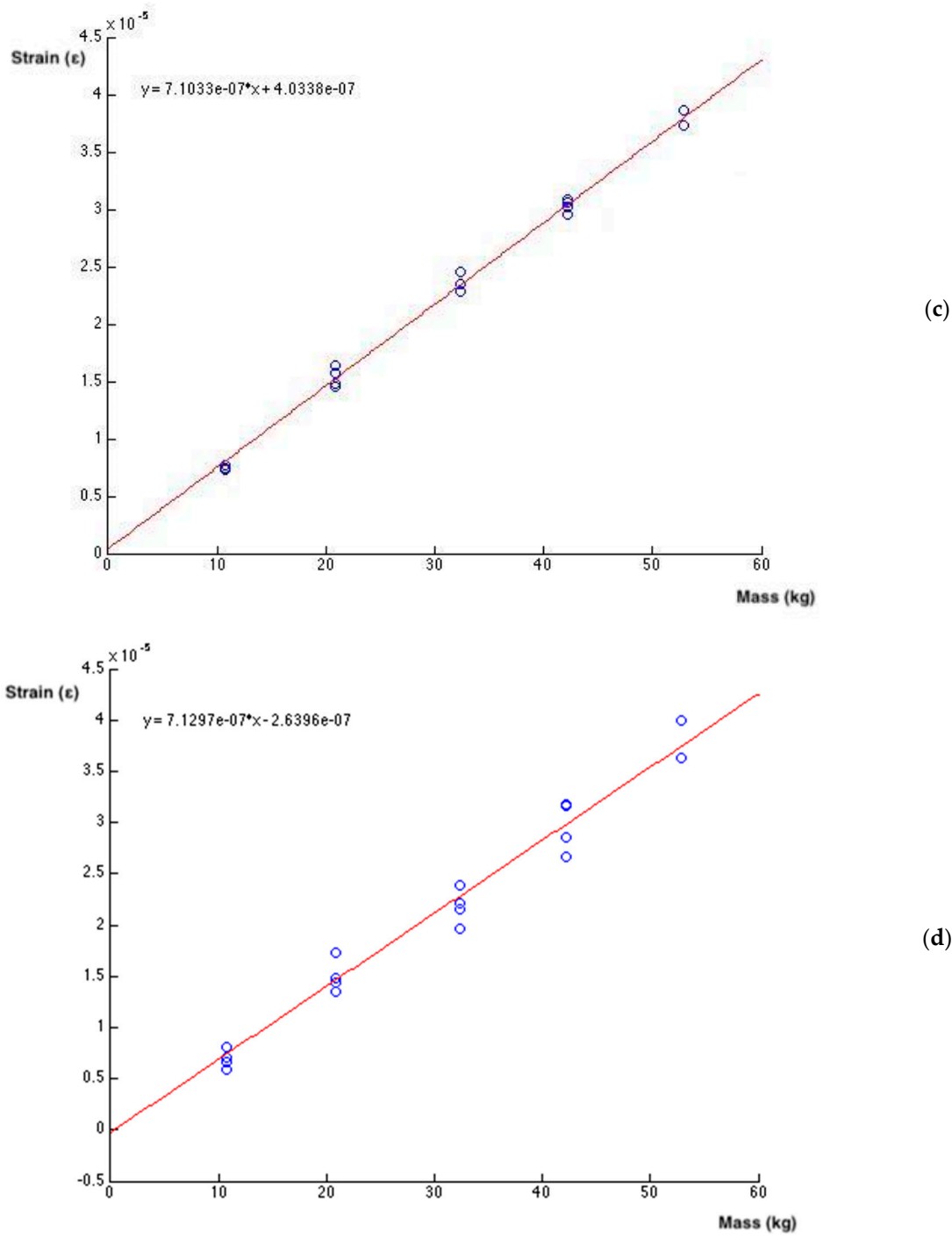

**Figure 5.** Force plates calibration results: 300 mm × 250 mm force plate antero-posterior force (**a** panel), medium-lateral force (**b** panel), 600 mm × 300 mm force plate antero-posterior force (**c** panel), and medium-lateral force (**d** panel).

Figure 6 presents vertical force, antero-posterior and lateral moment calibration (in normalised strain measure) for the underwater 600 mm × 300 mm force plate, evidencing linearization and cross-talk ~5% between orthogonal sensors (cf. [18,20]). The force plate design used has revealed greater sensibility in responses to the vertical loads, corroborating Lywood [21].

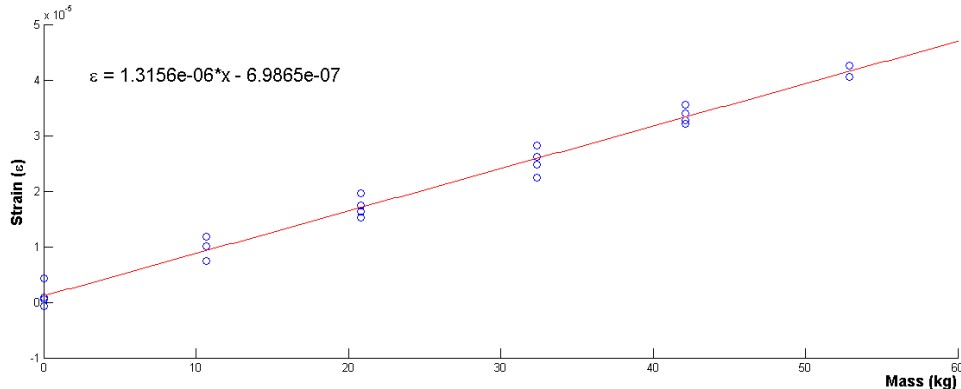

**Figure 6.** Force plates calibration results: 600 mm × 300 mm force plate vertical force.

Repeatability for loads above 300 N and below 40 N has revealed within 5% and 10–15% with a 95% confidence level. The $\overrightarrow{COP}$ locus was more uncertain farther from the platform centre with a reasonable radial around centre dependency.

### 3.3. Laboratory Experimental Validation

Due to in situ installation procedures, usage and aging, force plates accuracy may decrease, which can be propagated to the already calculated kinetic quantities. Based on these limitations, some research groups have developed systems to assess force plate accuracy using ad hoc designed in situ devices [8,26]. In the current study, static calibrations were followed by dynamical calibrations performed with a rigid body falling procedure [26], revealing homogeneity of static calibration results, particularly in the time = 0 force time curve of the falling body. The simultaneous correlation coefficient between the posterior strain filtered signal (moving average of 32 samples) and the theory generated by a previous application in MatLab R2014 (MathWorks Inc., Natick, MA,USA) has shown a value of 0.95.

### 3.4. Ecological Experimental Validation

The horizontal force–time curve exerted by hands and produced during a backstroke start data acquisition is similar to the one peak force profile previously presented (~0.5 BW; [24]). Most of the force–time curves displayed in backstroke start studies have analysed the horizontal component exerted on feet (e.g., [24,29]), which is similar to the dual peak force profile found in the current study (Figure 7a), with swimmers performing ~1.6 BW in the both peak forces. Vertical backstroke start forces exerted by feet have shown similar values to those found in de Jesus et al. (i.e., ~1.0 BW; [10,29]). Lateral force exerted by feet has depicted a peak force instant before the hands-off. Horizontal and vertical force–time curves exerted on feet and hands obtained in the new track start technique performed with the back plate (Figure 8) have registered similar rear (~800 N and ~1000 N) and front foot (~800 N and 600 N; cf. [1,8]) as well as hand force profiles (~150 N and ~800 N; cf. [8]). Previous studies have evidenced that the test–retest reliability of the kinetic gait parameters in the aquatic environment presented poor (medial lateral force component) to excellent reliability (vertical and antero-posterior force component; [30]).

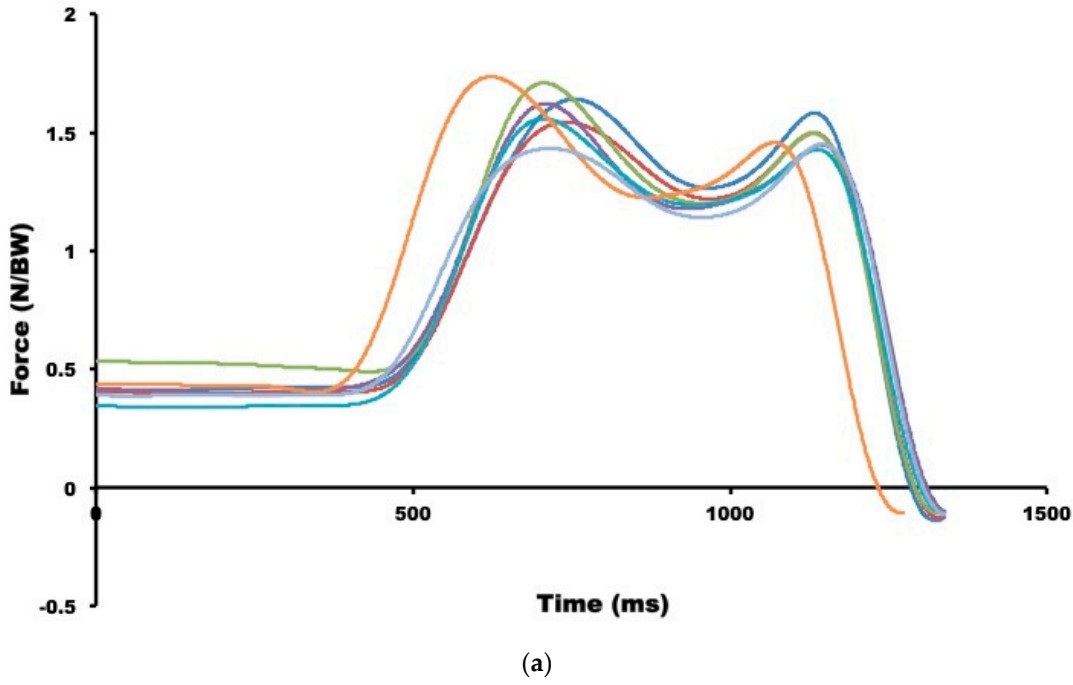

(**a**)

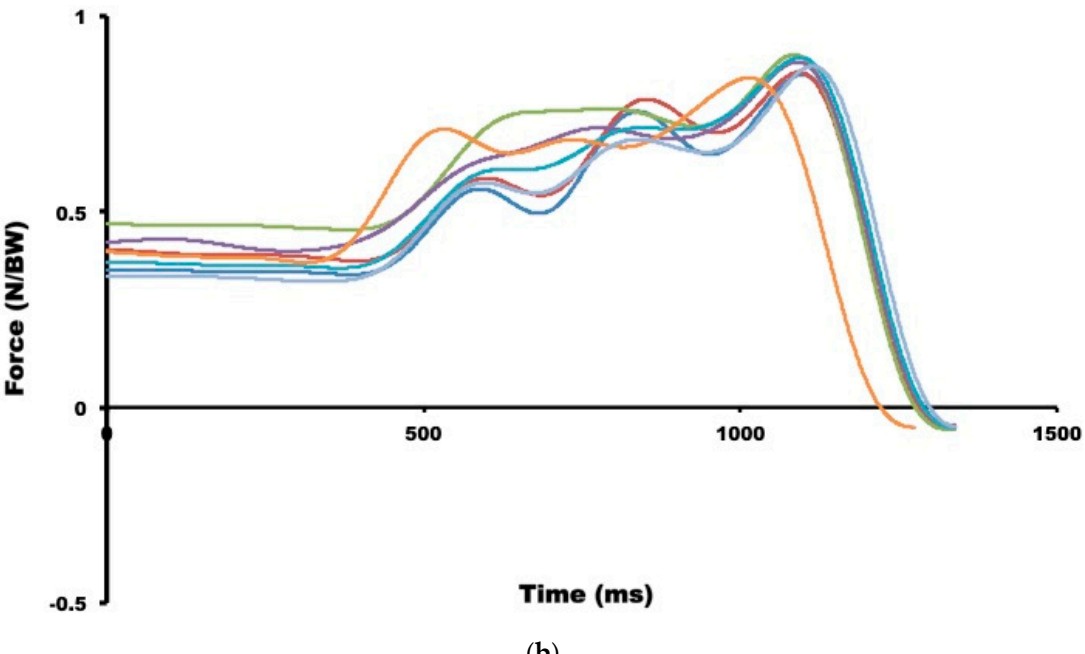

(**b**)

**Figure 7.** *Cont.*

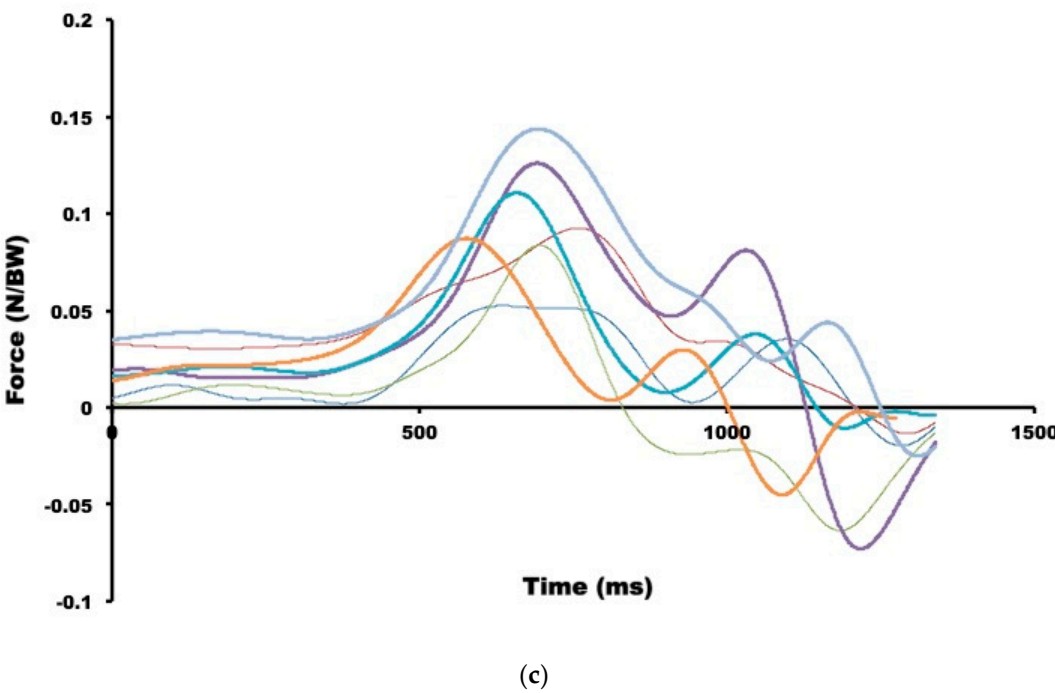

(**c**)

**Figure 7.** Backstroke start horizontal (**a** panel), vertical (**b** panel) and lateral (**c** panel) forces from seven trials of one swimmer (**c** panel).

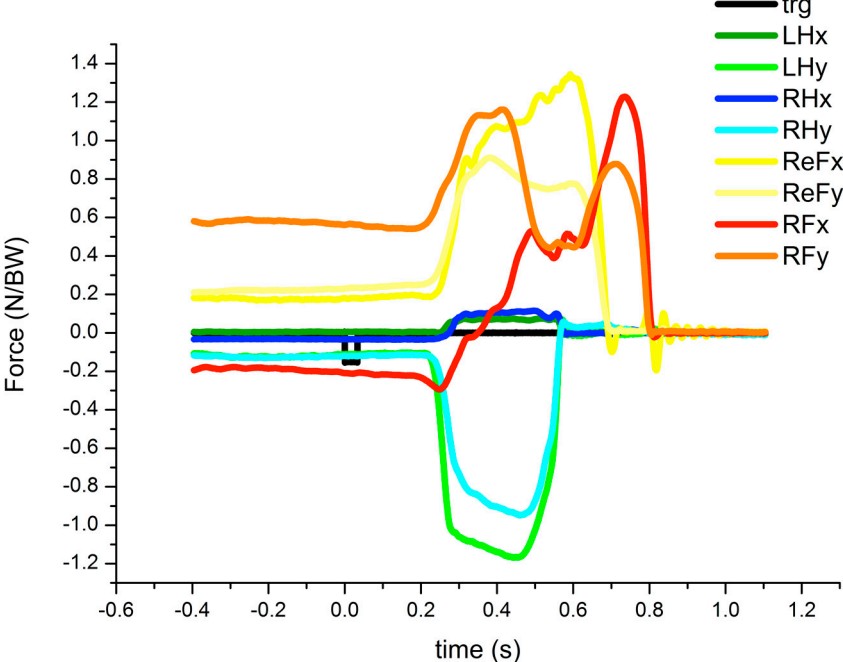

**Figure 8.** Individual antero-posterior (x) and vertical (y) force—time curves representing forces exerted on left and right hands (LH and RH), as well as rear and front foot (ReF and RF) in the new track start technique synchronised with the trigger (trg).

In relay start techniques, the horizontal force–time curve exerted on the feet and obtained during one step start technique (Figure 9) revealed a similar finding as obtained by Takeda et al. [31], although these authors had not measured right and left forces exerted on each foot.

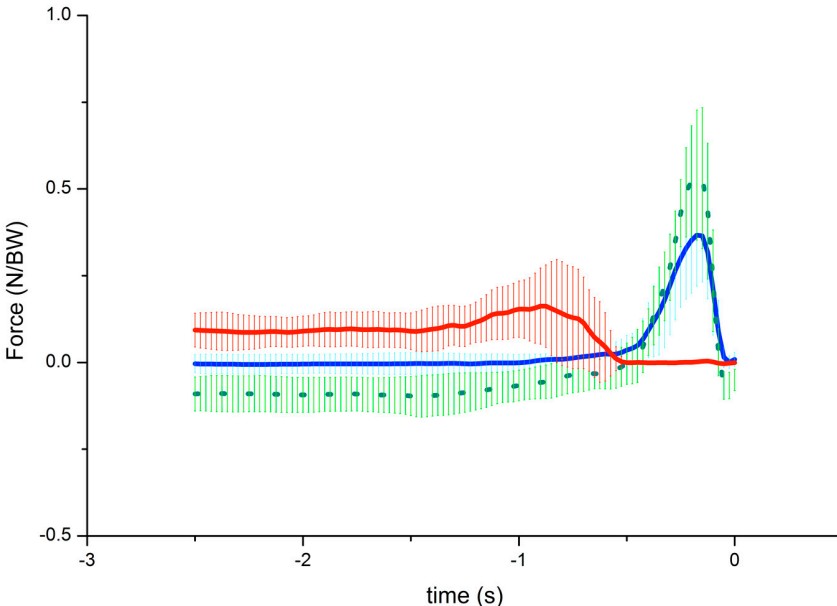

**Figure 9.** Horizontal force–time curve profiles from eight swimmers at the one step relay start technique: Rear foot (red line), front left and right foot (green and blue line, respectively).

Tri-axial forces measured in one underwater force plate during the open backstroke to breaststroke changeover have evidenced two distinct turn phases, the hand contact (~4.60 to 5.50 s) followed by swimmers' rotation (~4.80 s to 7.30 s) and the push-off (~7.30 to 7.80 s; Figure 10). Previous studies have already mentioned that the push-off phase depicted explosive lower limb movements since the first feet contact (~7.20 s) to the end of push-off (~7.80 s), which has been mainly observed in the horizontal force component [13,28]. Purdy and co-authors [31] have found ~193 N for the peak horizontal ground reaction force in female swimmers performing the open turn technique.

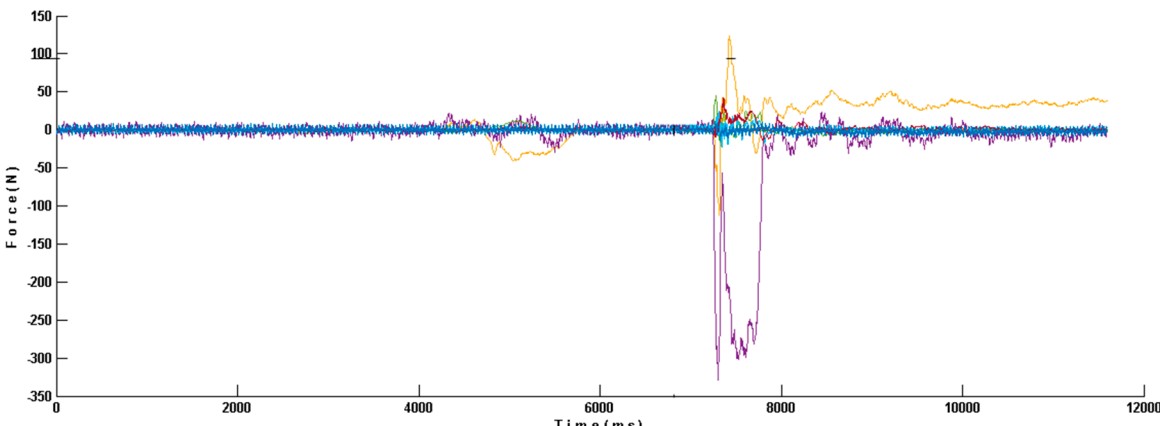

**Figure 10.** Individual backstroke to breaststroke open medley turning technique force–time curve with horizontal, vertical and medium-lateral components (purple, yellow and red line, respectively).

The current and the previously developed instrumented starting block configurations have shown restrictions regarding independent hand force and momentum measures with natural swimmers' hands and feet placement during all ventral start techniques. (e.g., [2,6,8]). The solution found in the current project was the replication of four more tri-axial force plates of 300 mm × 125 mm designed and numerically simulated (Figure 11a–d) to be placed in the frontal start block edge. An 800 N vertical, and 200 N medium-lateral and horizontal load were applied and registered 280.11, 109.89 and 117.34 με maximal deformation and a maximal 286.45 Hz frequency. The four-force plates solution

enables swimmers to grasp the top and bottom of the starting block surface, being versatile and valid to independently measure each swimmer's hand and foot forces and momenta generated regardless of the ventral starting technique used.

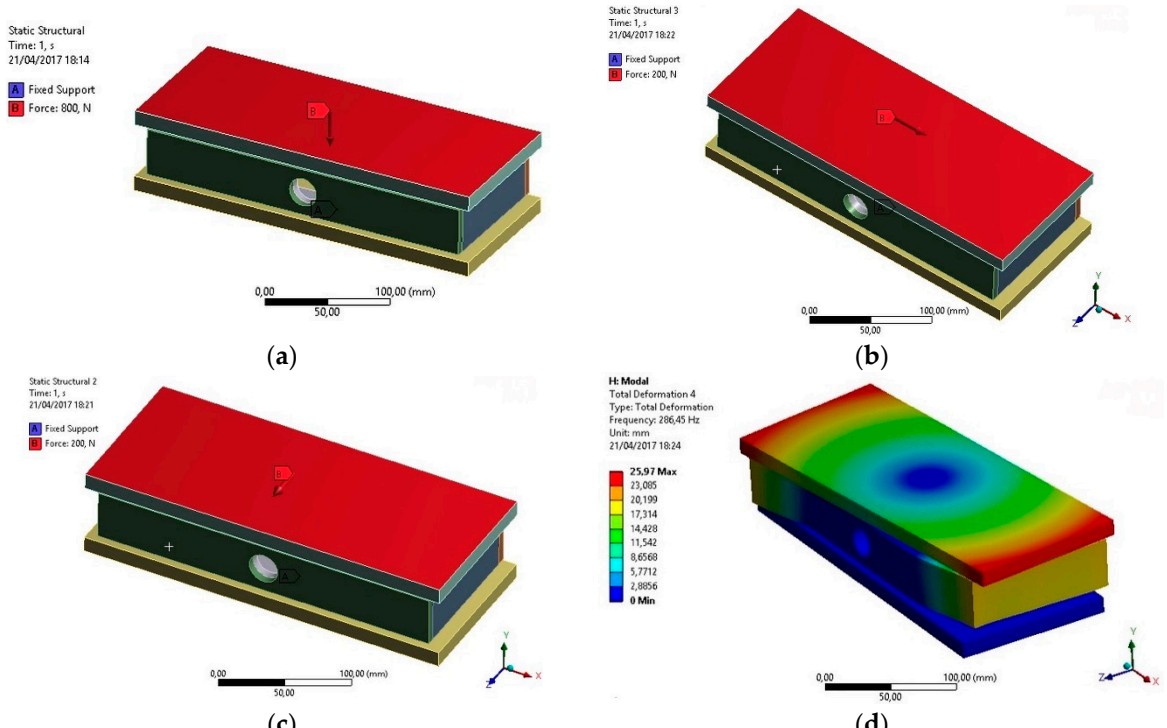

**Figure 11.** The four 300 mm × 125 mm force plates in static and dynamic simulation conditions for vertical, antero-posterior and medium lateral deformation (**a**–**c** panels), as well as the frequency analysis (**d** panel).

Notwithstanding the relevance of the dynamometric unit solution developed for start and turn analysis, it is recognised that future work is needed to improve main limitations: (i) finite element analysis revealed important data for project manufacturing. However, irregularities in the starting pool wall cannot be controlled and could affect measurements; (ii) despite static and dynamic calibration procedures, force plate linear responses were revealed, suggesting a less time-consuming in situ calibration procedure should be developed to avoid the detachment of each force plate from the dynamometric central; and (iii) increasing underwater force plate placement possibilities beyond side-by-side but also in an up and down location would cover independent feet force measurements from other turn techniques.

## 4. Conclusions

The dynamometric central for individual and relay swimming start and turn techniques framed according to FINA facility rules has evidenced reliable and accurate external force data. This device is ecologically valid and versatile, being able to be used in integrated or in future independent force plate analysis, as bow wave measurements in turns, passive drag and jumping techniques.

**Author Contributions:** Conceptualization, K.d.J. (Karla de Jesus), L.M., H.R., M.V. and J.P.V.-B.; data curation, K.d.J. (Karla de Jesus), L.M., M.V., R.F. and J.P.V.-B.; formal analysis, K.d.J. (Karla de Jesus), L.M., H.R., N.V., M.V. and J.P.V.-B.; funding acquisition, K.d.J. (Karla de Jesus), M.V., R.F. and J.P.V.-B.; investigation, K.d.J. (Karla de Jesus), K.d.J. (Kelly de Jesus), M.V., R.F., J.P.V.-B.; methodology, K.d.J. (Karla de Jesus), LM., H.R., N.V., K.d.J. (Kelly de Jesus), M.V., R.F., J.P.V.-B.; software, L.M., H.R., N.V., M.V., project administration; K.d.J. (Karla de Jesus), R.F., J.P.V.-B.; resources, K.d.J. (Karla de Jesus), R.F., J.P.V.-B.; supervision; K.d.J. (Karla de Jesus), L.M., M.V., R.F., J.P.V.-B.; validation; K.d.J. (Karla de Jesus), L.M., K.d.J. (Kelly de Jesus), J.P.V.-B.; visualization, K.d.J. (Karla de

Jesus), R.F., J.P.V.-B.; writing—original draft, K.d.J., (Karla de Jesus), L.M., H.R., N.V., K.d.J. (Kelly de Jesus), R.F., J.P.V.-B.; writing—review and editing, K.d.J. (Karla de Jesus), L.M., M.V., R.F., J.P.V.-B.

**Funding:** This research was funded by the [Coordination for the Improvement of Higher Education Personnel—CAPES] grant number [BEX 0761/12-5/2012-2015], [Foundation for Science and Technology—FCT] grant number [EXPL/DTP-DES/2481/2013-FCOMP-01-0124-FEDER-041981], [CAPES-FCT] grant number [99999.008578/2014-01], SANTADER grant number [PP/IJUP2011/123] and [Amazonas State Research Support Foundation - FAPEAM] grant number [POSGRAD 2017 FAPEAM, 002/2016]. Funders had no role in study design, data collection and analysis, decision to publish, or manuscript preparation.

**Acknowledgments:** Leandro dos Santos Coelho, Alexandre Igor Araripe Medeiros and Phornpot Chainok provided valuable feedback on manuscript drafts.

**Conflicts of Interest:** The authors declare no conflict of interest. The funders had no role in the design of the study; in the collection, analyses, or interpretation of data; in the writing of the manuscript, or in the decision to publish the results.

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
