# Peer review of "3D Device for Forces in Swimming Starts and Turns"

_applsci, doi:10.3390/app9173559_

Round 1
Reviewer 1 Report
[Major comments]
I would like to recognize the authors’ achievement of their newly developed force sensor. The original work of this 6DOF force sensor for start and turn will be fully welcomed and accepted by the swimmers and coaches. I believe that it must be beneficial for the performance enhancement in swimming.
However, the title should be changed. “3D-6DoF device for forces in swimming”. The reader might misunderstand your contents from the current title. In my understanding, of course, the start and turn might be included in the “swimming event”. However, the sensor itself is not for “swimming motion”. It only means that your developed equipment is for the start and turn motion in swimming. Also, 3D-6DoF must be corrected. 6DoF, six degree of freedom does mean already three-dimensional application.
What I think the authors’ original paper ends with a somewhat disappointing result is that the main story of this manuscript is a development report of the force sensor. From the view point of the engineering, it is quite simple to disassemble the starting block sensor into three parts. Also, as for the vertical force sensor for the backstroke start with wedges is a basic strategy to develop. I think this manuscript is very informative, but I could not find new findings and innovative strategy to develop.
After explanation of the sensor specification and its mechanical properties with some simulation studies, the authors introduced only one single result of the experimental data (Fig. 7). In the section of 2.9, Ecological experimental validation, the author mentioned that they collected number of swimmers (adults and age-grouped) and tested. However, I think that one single figure does not support for this number of subjects’ validation experiment. Therefore, I reviewer recommend the author should submit this manuscript as the technical report not an original paper.
[Minor comments]
P1L37. R2 should be corrected to R2. (R squared) as mathematical formulation.
P10 to P12 Figure 5. The unit of the vertical axis is wrong. The vertical axis has the unit of (micro strain). However, the numerical demonstration of the axis itselt is “x 10-5” etc. There must be the mismatch of those expressions. In addition, those plot of the data with estimation equation might be direct output by some software. The obtained data point is difficult to recognize and the numbers of horizontal and vertical axis steps are too small to identify. Please replace all of graph with more reader-friendly plots.
P13 Figure 6.
Figure 6(b) is explained as the “moment” calibration. However, the unit of the horizontal axis of Figure 6(b) is “Force[N]”. I could not understand this explanation and the graph.
P14 Figure 7.
Figure 7 is too complicated to understand for the readers. Different axial results of different trials are on the same graph window. Please replace Figure 7 with more reader friendly graphs. I think that the graph should be divided into three graphs, such as the vertical, anterior-posterior and medio-lateral direction, for example.
P15 Figure 9.
The authors explain that Figure 9 is the result of step relay start. Then, the caption of Figure 9 says that “front left and right foot (green and blue line). However, on the Figure 9, the green line is the vertical which might be expressed for the standard deviation (or 95% confident level?). There are green dots on each vertical green line. But it differs with other red and blue lines. Do I make a misunderstanding for Figure 9?
P16 Figure 10
It is too ugly plot graph. Please replace and make an amendment for the graph with visible labels. I could see its label of horizontal axis “Time(s)”. Is it true? Between 7200 to 7900 (roughly estimated), the author explained it as the turn. But, the graph shows its duration is about 700(sec). Please correct this figure.
From P16, in the references.
There are many references which does not have its publish year.
References with number 2,5,11,16,17,19 and 27 do not have year.
Author Response
Reviewer# 1: I would like to recognize the authors’ achievement of their newly developed force sensor. The original work of this 6DOF force sensor for start and turn will be fully welcomed and accepted by the swimmers and coaches. I believe that it must be beneficial for the performance enhancement in swimming.
Authors:We thank you for reading our manuscript and providing a positive overview.
Reviewer# 1: However, the title should be changed. “3D-6DoF device for forces in swimming”. The reader might misunderstand your contents from the current title. In my understanding, of course, the start and turn might be included in the “swimming event”. However, the sensor itself is not for “swimming motion”. It only means that your developed equipment is for the start and turn motion in swimming. Also, 3D-6DoF must be corrected. 6DoF, six degree of freedom does mean already three-dimensional application.
Authors:We have corrected accordingly.
Reviewer# 1: What I think the authors’ original paper ends with a somewhat disappointing result is that the main story of this manuscript is a development report of the force sensor. From the view point of the engineering, it is quite simple to disassemble the starting block sensor into three parts. Also, as for the vertical force sensor for the backstroke start with wedges is a basic strategy to develop. I think this manuscript is very informative, but I could not find new findings and innovative strategy to develop.
Authors:What is not simple is to split in a minimum of 7 platforms addressing each limb contact in all FINA’s allowed ventral and dorsal individual and relay start techniques body segment signature, laterality effects and turn analysis. Therefore, the main originality is the universality available.
Reviewer# 1:After explanation of the sensor specification and its mechanical properties with some simulation studies, the authors introduced only one single result of the experimental data (Fig. 7). In the section of 2.9, Ecological experimental validation, the author mentioned that they collected number of swimmers (adults and age-grouped) and tested. However, I think that one single figure does not support for this number of subjects’ validation experiment. Therefore, I reviewer recommend the author should submit this manuscript as the technical report not an original paper.
Authors: We have included more than one figure, considering swimming event components studied with the presents instrumented starting block: (i) individual ventral starts – kick start example, (ii) relay start – one step technique (eight swimmers), (iii) backstroke start (seven trials of the same swimmer), (iv) backstroke to breaststroke medley turn (one individual example). Following the manuscript guidelines for conciseness, authors understand that the most important results have been included in this original research.
Reviewer# 1:P1L37. R2 should be corrected to R2. (R squared) as mathematical formulation.
Authors: We have corrected accordingly.
Reviewer# 1: P10 to P12 Figure 5. The unit of the vertical axis is wrong. The vertical axis has the unit of (micro strain). However, the numerical demonstration of the axis itselt is “x 10-5” etc. There must be the mismatch of those expressions. In addition, those plot of the data with estimation equation might be direct output by some software. The obtained data point is difficult to recognize and the numbers of horizontal and vertical axis steps are too small to identify. Please replace all of graph with more reader-friendly plots.
Authors: We have improved Figure 5 accordingly
Reviewer# 1: Figure 6 (b) is explained as the “moment” calibration. However, the unit of the horizontal axis of Figure 6(b) is “Force[N]”. I could not understand this explanation and the graph.
Authors: We have corrected accordingly
Reviewer# 1: Figure 7 is too complicated to understand for the readers. Different axial results of different trials are on the same graph window. Please replace Figure 7 with more reader friendly graphs. I think that the graph should be divided into three graphs, such as the vertical, anterior-posterior and medio-lateral direction, for example.
Authors: For sake of better track start movements sequence visualisation from each limb and component we have decided to keep force-time curves in the same graph, but we have improved image quality, normalised force data to the body weight and added a legend.
Reviewer# 1: The authors explain that Figure 9 is the result of step relay start. Then, the caption of Figure 9 says that “front left and right foot (green and blue line). However, on the Figure 9, the green line is the vertical which might be expressed for the standard deviation (or 95% confident level?). There are green dots on each vertical green line. But it differs with other red and blue lines. Do I make a misunderstanding for Figure 9?.
Authors: The three data points sequence is from the anterior-posterior force. Vertical force during the one step relay start technique is not represented in the manuscript. The red is the rear foot, blue and green the right and left foot. Data presented are in accordance with Takeda’s et al. 2010 paper.
Reviewer# 1: It is too ugly plot graph. Please replace and make an amendment for the graph with visible labels. I could see its label of horizontal axis “Time(s)”. Is it true? Between 7200 to 7900 (roughly estimated), the author explained it as the turn. But, the graph shows its duration is about 700(sec). Please correct this figure
Authors: We have corrected accordingly
Reviewer# 1: There are many references which does not have its publish year.
References with number 2,5,11,16,17,19 and 27 do not have year
Authors: We have corrected accordingly.

Reviewer 2 Report
In my opinion, this article can be acceptable for publication. Current research problem is interesting for practice. The paper has originality, rationality and completeness of research problem, originality of data, good tables and figures, good discussion and explanation of findings. It is clearly written. The methods are appropriate. The conclusions are reasonable. Writing style and language quality is at the academic level.
Author Response
Reviewer# 2: In my opinion, this article can be acceptable for publication. Current research problem is interesting for practice. The paper has originality, rationality and completeness of research problem, originality of data, good tables and figures, good discussion and explanation of findings. It is clearly written. The methods are appropriate. The conclusions are reasonable. Writing style and language quality is at the academic level.
Authors: We appreciate the very positive comments from the Reviewer.

Reviewer 3 Report
Dear author,
Thank you for presenting your research on the development of an instrumented start block and wall for the kinetic analysis of swimming starts and turns. New technologies to support practitioners and researchers in this area are always of interest when offering advantages over currently available systems.
The system you are presenting is as such of interest as it outlines 2 major improvements compared to other systems (separate force measurements for each limb and cost effectiveness). However, in my opinion these points aren’t adequately addressed throughout the paper. Nor is the validation presented in this paper convincing enough for end-users of the system. Therefore, I feel like the paper needs to be improved on several aspects before being viable for publication.
I also encourage you to seek language advice to improve readability and conciseness (e.g. 30-31 remove ‘using kinetic analysis’, 40 remove ‘positioned’) of your writing. This will help in getting your message across in a cleaner and more transparent way.
More specific comments on the paper can be found below.
I am willing to review this paper again once all comments have been answered/addressed.
Best regards.
Introduction:
The very first statement in the abstract refers to systems similar to the one presented in this article having been developed ever since 1970. Therefore, in my opinion, the introduction should include an overview of the systems that already exist. Several widely known systems (Wetplate, Kistler, Bertec, …) aren’t mentioned in the introduction. Furthermore, the introduction should also identify the shortcomings of those systems and highlight the areas that the currently developed system addresses, and how it as such is a valuable contribution to the already available technologies. These areas are quickly mentioned and include:
- The ability to measure forces separately for each limb, and this as well above-water as below-water. It would be good to make the reader understand where swimmers generate forces on the block/wall during a start/turn; and how these then could be measured. A basic movement description of widely used starting/turning techniques would offer more insight into suitability of specific force platform placement for biomechanical analysis. (e.g. where are the feet placed during tumble turns (and is the side-to-side placement of the force platforms therefore the optimal solution ?, …)
- Where this system surpasses other systems is the separate measurement for both hands. It would be good to get a better understanding as to why the authors consider this capability important (especially given that the arm action in swim starts rarely deviates from a symmetrical movement).
- The authors claim a lower cost than other systems. This point should also be properly discussed.
I feel like certain sections of the current introduction could be omitted (e.g. the classification of starts and turns) in favour of addressing the above.
Methods:
Avoid repetition of titles (e.g. 2.5 = 2.6)
110 - What do the authors mean by ‘time layout’ ?
194 - Gravity at sea level is usually rounded to 9.81 m/s².
245 - Static calibration. Why did you limit static calibration to 50 kg and does this cover the full range needed for the purpose ? e.g. it is known that during single leg push-off in a track start resultant forces rise above 1* BW, which is for the majority of athletes well above 50 kg.
258 – centre of pressure : wouldn’t it be more correct to talk about ‘centre of force’ rather than ‘centre of pressure’ as you are technically measuring force and not pressure ?
266: why did you limit yourself to a qualitative and not a quantitative comparison ?
281: is it a back to breast turn (medley) or is it a backstroke and a breaststroke turn ?
Results & Discussion:
Table 1: what is the relevance of presenting data of first and third evolutions of the system ? For most end-users the final prototype is the only one of interest.
Table 2: this table needs more explanation in order to properly interpret results. Eg. Numbering of the strain gauges and what they are measuring. Would there be more visually intuitive ways of presenting this type of data ?
Table 2: it would be interesting to get a full understanding of cross-talk. Would it be beneficial to run an analysis of how each direction influences another instead of just the influence of the vertical onto the others ?
Figure 6 : I don’t understand figure 6. Please explain the procedure of testing used. Why isn’t there a linear relationship between force applied and strain ?
3.3 and 3.4 have the same title ?
3.3 does not present or discuss results
3.4 – the methods used for the ecological validation seem inadequate. I would like to get more quantitative data comparing to quantitative measures in literature. Furthermore, I would like to see figure 7 expressed in body weight. This makes it possible to check simple Newtonian mechanics. A legend is needed in order to understand which traces are which signals.
391-398: I don’t understand how this ties into the current system. Can you provide some more practical information on how these components are integrated ?
Figure 9: is the unit for force measurement correct ? (N/BW)
417 : it is more easy to relate to these results when expressing timings relative to first contact instead of time of start of measurement ?
Figure 10: are the presented forces correct in this figure ? The max force seems quite low for a turn performance.
End-users of this system will be interested in the accuracy of the system. I feel like it would be very worthwhile adding a results section devoted to this.
It is also useful to know the limitations of this system.
- E.g. the platforms are mounted side-by-side on the wall. Could the system be adapted to also measure turns in which the feet are placed above eachother ?
- What about the measurement of incoming water flow ? As the athlete swims into the wall, an important amount of water is also moving towards the wall. The solid surface of the turn wall as such not only measures reaction forces relating to the swimmers’ actions, but also water flow. Other systems (e.g. Kistler) provide perforations to minimize this effect. How big would this influence be on your system ?
Conclusions:
It is claimed that the device can be used for wave drag measurement, however, no evidence has been given in the paper to support this claim.
Author Response
Reviewer# 3: Thank you for presenting your research on the development of an instrumented start block and wall for the kinetic analysis of swimming starts and turns. New technologies to support practitioners and researchers in this area are always of interest when offering advantages over currently available systems
Authors: We appreciate the very positive overview provided
Reviewer# 3: The system you are presenting is as such of interest as it outlines 2 major improvements compared to other systems (separate force measurements for each limb and cost effectiveness). However, in my opinion these points aren’t adequately addressed throughout the paper. Nor is the validation presented in this paper convincing enough for end-users of the system. Therefore, I feel like the paper needs to be improved on several aspects before being viable for publication
Authors: We thank the reviewer for their careful reading of the manuscript and their constructive remarks. We have taken the comments on board to improve and clarify the manuscript.
Reviewer# 3: I also encourage you to seek language advice to improve readability and conciseness (e.g. 30-31 remove ‘using kinetic analysis’, 40 remove ‘positioned’) of your writing. This will help in getting your message across in a cleaner and more transparent way.More specific comments on the paper can be found below. I am willing to review this paper again once all comments have been answered/addressed.
Best regards
Authors: We have improved language accordingly.
Reviewer # 3: The very first statement in the abstract refers to systems similar to the one presented in this article having been developed ever since 1970. Therefore, in my opinion, the introduction should include an overview of the systems that already exist. Several widely known systems (Wetplate, Kistler, Bertec, …) aren’t mentioned in the introduction. Furthermore, the introduction should also identify the shortcomings of those systems and highlight the areas that the currently developed system addresses, and how it as such is a valuable contribution to the already available technologies.
Authors: We have added an overview of the instrumented starting block already available accordingly.
Reviewer # 3: The ability to measure forces separately for each limb, and this as well above-water as below-water. It would be good to make the reader understand where swimmers generate forces on the block/wall during a start/turn; and how these then could be measured. A basic movement description of widely used starting/turning techniques would offer more insight into suitability of specific force platform placement for biomechanical analysis. (e.g. where are the feet placed during tumble turns (and is the side-to-side placement of the force platforms therefore the optimal solution ?, …)
Authors: We have added more insight into start and turns description and force plates location.
Reviewer # 3: Where this system surpasses other systems is the separate measurement for both hands. It would be good to get a better understanding as to why the authors consider this capability important (especially given that the arm action in swim starts rarely deviates from a symmetrical movement).
Authors: We have improved the instrumented starting block potentials’ descritption accordingly.
Reviewer # 3: The authors claim a lower cost than other systems. This point should also be properly discussed.
Authors: We have improved force plate costs discussion.
Reviewer # 3: I feel like certain sections of the current introduction could be omitted (e.g. the classification of starts and turns) in favour of addressing the above
Authors: We have corrected accordingly.
Reviewer # 3: Avoid repetition of titles (e.g. 2.5 = 2.6)
Authors: We have corrected accordingly.
Reviewer # 3: What do the authors mean by ‘time layout’ ?
Authors: We have removed the word “time”.
Reviewer # 3: Gravity at sea level is usually rounded to 9.81 m/s².
Authors: We have corrected accordingly
Reviewer # 3: Static calibration. Why did you limit static calibration to 50 kg and does this cover the full range needed for the purpose? e.g. it is known that during single leg push-off in a track start resultant forces rise above 1* BW, which is for the majority of athletes well above 50 kg.
Authors: Our research group has previously used higher values during calibration procedures in similar force plate design (c.f. Mourão et al. 2015) and the multiplicative factor was similar. However, it is understandable to discourage extrapolation. Future works could include higher force static calibration. Previous studies have also used similar total masses number to calibrate load cells that where used in swimming starts (e.g. Takeda et al., 2017).
Reviewer # 3: centre of pressure : wouldn’t it be more correct to talk about ‘centre of force’ rather than ‘centre of pressure’ as you are technically measuring force and not pressure ?
Authors;Indeed momentum is measured and gives limb COP. For language homogeneity adoption of the same term seemed more coherent.
Reviewer # 3:why did you limit yourself to a qualitative and not a quantitative comparison?
Authors: We have added previous literature data in the Discuss to improved quantitative visualisation.
Reviewer # 3:is it a back to breast turn (medley) or is it a backstroke and a breaststroke turn ?
Authors: We have added “medley” to clarify.
Reviewer # 3:Table 1: what is the relevance of presenting data of first and third evolutions of the system? For most end-users the final prototype is the only one of interest.
Authors: The authors understand that the force plates design steps can evidence quantitatively the results from the compromise among maximum rigidity, minimum mass and high frequency. We have better described this compromise in the item 3.1.
Reviewer # 3:This table needs more explanation in order to properly interpret results. Eg. Numbering of the strain gauges and what they are measuring. Would there be more visually intuitive ways of presenting this type of data?
Authors: We have improved the Table 2 description and included references for each strain gauge.
Reviewer # 3:I don’t understand figure 6. Please explain the procedure of testing used. Why isn’t there a linear relationship between force applied and strain ?
Authors: We have corrected accordingly.
Reviewer # 3: 3.3 and 3.4 have the same title ?
Authors: We have corrected accordingly.
Reviewer # 3: 3.3 does not present or discuss results
Authors: We have improved this sentence accordingly.
Reviewer # 3:The methods used for the ecological validation seem inadequate. I would like to get more quantitative data comparing to quantitative measures in literature. Furthermore, I would like to see figure 7 expressed in body weight. This makes it possible to check simple Newtonian mechanics. A legend is needed in order to understand which traces are which signals.
Authors: We have added quantitative for better visualisation data of obtained results. We have improved image quality, normalised force data to the body weight and included a legend.
Reviewer # 3:I don’t understand how this ties into the current system. Can you provide some more practical information on how these components are integrated ?
Authors: We have better described in the manuscript 3.4 section.
Reviewer # 3:Figure 9: is the unit for force measurement correct ? (N/BW).
Authors: As in previous papers, the unit N/BW has been presented to show the forces as a fraction of body weight (e.g. Nguyen et al., 2014;Barkwell et al., 2017; Pereira et al., 2015).
Reviewer # 3:417: it is more easy to relate to these results when expressing timings relative to first contact instead of time of start of measurement?
Authors: We understand the Reviewer’s suggestion. However, we have adopted a similar time graph arrangement to the previous paper from Takeda et al., 2010 to facilitate data visualisation and discussion.
Reviewer # 3:417: Figure 10: are the presented forces correct in this figure? The max force seems quite low for a turn performance.
Authors: The data presented in Figure 10 is representative of only of the underwater force plates and values are close to previous studies (e.g. Purdy et al., 2012). We have included this reference in the Results and Discussion section to improve data Discussion.
Reviewer # 3:417: End-users of this system will be interested in the accuracy of the system. I feel like it would be very worthwhile adding a results section devoted to this.
Authors: We have added a sentence in the Results and Discussion section accordingly.
Reviewer # 3:It is also useful to know the limitations of this system.
Authors: We have improved the study limitations description in Conclusion section paragraph and removed these sentences to the end of the Results and Discussion section accordingly.
Reviewer # 3:What about the measurement of incoming water flow ? As the athlete swims into the wall, an important amount of water is also moving towards the wall. The solid surface of the turn wall as such not only measures reaction forces relating to the swimmers’ actions, but also water flow. Other systems (e.g. Kistler) provide perforations to minimize this effect. How big would this influence be on your system ?
Authors: Despite the Kistler’s solution is user friendly, our research group would like to continue characterizing and quantifying the bow wave that precedes the contact of the swimmer with the wall in different turn techniques, since we have previously used a graphical removal technique on turns (Pereira et al., 2010). Furthermore, dynamometric unit force plates can be used for other analysis purposes, which do not include bow wave effects.
Reviewer # 3: It is claimed that the device can be used for wave drag measurement, however, no evidence has been given in the paper to support this claim
Authors: We have better described this sentence. Unfortunately, we were not able to measure passive drag, but an accurate force plate can be adapted to be used as a sensor for towing devices, as previously adapted with a waterproof Kistler force plate (e.g Formosa et al., 2011). Considering the water wave impact, we will present in future experimental analysis using the current system.

Reviewer 4 Report
This paper is the report of development of a device to measure forces in swimming. The developed device is sufficiently described. However, in the development itself, no scientific significance can be found. This device is basically a very classical strain gauge type force plate. The finite element analysis also does not have anything novel from the scientific viewpoint.
The reviewer is personally very looking forward to further researches by the authors regarding to the mechanics of swimming, using the developed device.
However, the reviewer cannot recognize sufficient scientific value in this report.
Author Response
Reply to Reviewer’s # 4 comments
Reviewer # 4: This paper is the report of development of a device to measure forces in swimming. The developed device is sufficiently described. However, in the development itself, no scientific significance can be found. This device is basically a very classical strain gauge type force plate. The finite element analysis also does not have anything novel from the scientific viewpoint.
Authors: Thank you for reading our manuscript. As already presented to Reviewer’s # 1, what is not simple is to split in a minimum of 7 platforms addressing each limb contact in all FINA’s allowed ventral and dorsal individual and relay start techniques body segment signature, laterality effects and turns. Therefore, the main originality is the universality available.
Reviewer # 4: The reviewer is personally very looking forward to further researches by the authors regarding to the mechanics of swimming, using the developed device.
Authors: We thank you for reading our manuscript and providing a positive overview
Reviewer # 4: However, the reviewer cannot recognize sufficient scientific value in this report
Authors: We sincerely appreciate all valuable comments and suggestions from the four reviewers, which helped us to improve the quality of the manuscript.

Round 2
Reviewer 4 Report
Thank you for the revision. Since many parts were much improved, now the paper is worth to be published.